**Data Availability Statement:** All relevant data are within the manuscript and its Supporting information files.

**Funding:** This research study was unfunded.

# Growth and the pubertal growth spurt in South African adolescents living with perinatally-acquired HIV infection

**Bilema Mwambenu**[1]*, **Vundli Ramoloko**[2], **Ria Laubscher**[3], **Ute Feucht**[1,4,5,6]

1 Department of Paediatrics, Kalafong Hospital, University of Pretoria, Pretoria, South Africa, 2 Health Systems Research Unit, South Africa Medical Research Council, Cape Town, South Africa, 3 Biostatistics Unit, South Africa Medical Research Council, Cape Town, South Africa, 4 Research Centre for Maternal, Fetal, Newborn and Child Health Care Strategies, University of Pretoria, Pretoria, South Africa, 5 Maternal and Infant Health Care Strategies Research Unit, South African Medical Research Council, Pretoria, South Africa, 6 Gauteng Department of Health, Tshwane District Health Services, Johannesburg, South Africa

* marthymwambenu@gmail.com

## Abstract

### Background

The majority children living with HIV infection now survive into adulthood because of effective antiretroviral therapy (ART), but few data exist on their growth during adolescent years. This study investigated growth patterns and evaluated factors associated with suboptimal growth in adolescents with perinatally-acquired HIV infection.

### Methods

This retrospective cohort study included HIV-infected adolescents, aged 13 to 18 years, with at least 5 years of ART follow-up at a large HIV clinic in the Gauteng Province, South Africa. Weight-for-age Z-scores (WAZ), height-for-age Z-scores (HAZ) and body mass index (BMI)-for-age Z-scores were calculated using World Health Organization (WHO) growth standards. Growth velocity graphs were generated utilising the mean height change calculated at 6-monthly intervals, using all available data after ART initiation, to calculate the annual change. Other collected data included WHO HIV disease staging, CD4%, HIV viral loads (VLs), ART regimens and tuberculosis co-infection.

### Results

Included were 288 children with a median age of 6.5 years (IQR 4.2;8.6 years) at ART initiation, and 51.7% were male. At baseline the majority of children had severe disease (92% WHO stages 3&4) and were started on non-nucleoside reverse transcriptase inhibitor-based regimens (79.2%). The median CD4% was 13.5% (IQR 7.9;18.9) and median HIV viral load log 5.0 (IQR 4.4;5.5). Baseline stunting (HAZ <-2) was prevalent (55.9%), with a median HAZ of -2.2 (IQR -3.1;-1.3). The median WAZ was -1.5 (IQR -2.5;-0.8), with 29.2% being underweight-for-age (WAZ <-2). The peak height velocity (PHV) in adolescents with baseline stage 3 disease was higher than for those with stage 4 disease. Being older at ART

**Competing interests:** The authors have declared that no competing interests exist.

start (p<0.001) and baseline stunting (p<0.001) were associated with poorer growth, resulting in a lower HAZ at study exit, with boys more significantly affected than girls (p<0.001).

## Conclusions

Suboptimal growth in adolescents with perinatally-acquired HIV infection is a significant health concern, especially in children who started ART later in terms of age and who had baseline stunting and is more pronounced in boys than in girls.

## Introduction

Children living with perinatally-acquired HIV infection are now surviving into adulthood in large numbers, as HIV infection has become a treatable chronic condition due to the availability of antiretroviral therapy (ART) [1–3]. This has resulted in adolescent HIV infection becoming an ever-increasing health care burden, especially in high-burden countries. According to reports about 2 million adolescents between the ages of 10 and 19 years are living with HIV infection worldwide, with the majority in Sub-Saharan Africa (89%) [4, 5]. Barriers to successful treatment include the need for lifelong therapy, denial of the diagnosis, and limited understanding of treatment benefits and treatment side effects such as lipodystrophy and gynaecomastia, which can lead to stigmatisation [6]. These factors may lead to poor treatment adherence with resultant development of viral resistance and other health issues, including delayed pubertal growth and poor height growth velocity [6].

Previous studies, mainly from high-income settings, have documented an association between paediatric HIV infection and growth failure as well delayed pubertal onset [7–11]. Implicated factors included HIV-related opportunistic infections, underlying malnutrition and chronic inflammation [8, 12]. Data on the prevalence and severity of delayed pubertal growth and stunting in HIV-infected adolescents in Sub-Saharan Africa are limited, most of the studies were done in younger children, as reported in a study done by Feucht *et al.* in South Africa, where 20.2% of these children remain stunted after five years of ART [13]. This current study focused on factors associated with growth failure and a delayed and reduced pubertal growth spurt in adolescents living with perinatally-acquired HIV infection.

## Material and methods

This retrospective cohort study was conducted at the paediatric HIV clinic at the Kalafong Provincial Tertiary Hospital situated in the City of Tshwane, Gauteng Province, South Africa. The study population included all adolescents aged between 13 and 18 years with perinatally-acquired HIV infection who initiated ART before 12 years of age and had ART follow-up data for at least 5 years. Data were prospectively documented in the clinic's electronic database during routine clinical visits (September 2003 to May 2016) and retrospectively analysed. This included anthropometric measurements, HIV clinical disease severity (World Health Organization (WHO) classification [14]), tuberculosis co-infection, laboratory data (CD4 and HIV viral load (VL)), and level of treatment adherence (by clinical questionnaire and pill count).

The children were started on ART according to the South African National HIV guidelines at the time [15]. Children below 3 years of age were started on 2 nucleoside reverse transcriptase inhibitors (NRTIs) and a protease inhibitor (PI), and those above 3 years and 10 kg were initiated on a non-nucleoside reverse transcriptase inhibitor (NNRTI) instead of the PI. The

diagnosis of tuberculosis was made clinically, assisted with radiological and microbiological evidence.

Anthropometric measurements were taken every 4 to 12 weeks by registered dieticians during routine follow-up visits, and subsequently age- and sex-standardised using the WHO growth reference Z-scores [16]. Underweight was defined as a weight-for-age Z-score (WAZ) of ≤-2 to <-3, and severe underweight as WAZ ≤-3. Moderate stunting was defined as a height-for-age Z-score (HAZ) of ≤-2 to <-3, and severe stunting as HAZ ≤-3. Moderate wasting was defined as a body mass index (BMI)-for-age Z-scores (BAZ) of ≤-2 to <-3, and severe wasting as BAZ ≤-3. Overweight was defined as a BAZ between 2 and 3. Obesity was defined as a BAZ above 3 SD. Age at peak height velocity (PHV), which is the age of the maximum rate of growth during a growth spurt, was also assessed by studying the linear growth over time for participants who had follow-up until at least 16 years of age.

## Statistical analysis

Differences between males and females at baseline and study exit were assessed by chi-squared test (categorical variables) or Wilcoxon's Sum rank test (continuous variables). Statistical significance was indicated by a p-value <0.05. For the growth velocity graphs, the mean height was calculated at 6-monthly intervals, using all data after ART initiation. These mean heights were used to calculate the annual change. To smooth the graphs, a 2.5 years' moving average was calculated at the midpoint of the period. These were plotted against the WHO reference values for mean height for males and females in the corresponding age period. To graph the HAZ data, the mean HAZ was calculated at 6-monthly intervals for males and females. These graphs were then smoothed with a 2.5 years' moving average calculated at midpoint of the period. A generalised linear model was fitted with HAZ at study exit the outcome variable and age at ART initiation (ART initiation <6 years as reference level), baseline HAZ (extremely stunted as reference level), WHO staging (WHO stages 1 & 2 as reference level), tuberculosis co-infection (no tuberculosis as reference level), ART adherence (good adherence as reference level), sex (male as reference level), HIV viral load and degree of viral suppression (non-suppression as reference level) as predictors. Variables that were not statistically significant were excluded from the final predictive model.

The study was approved by the Faculty of Health Sciences Research Ethics Committee of the University of Pretoria, as well as obtaining the relevant institutional permissions. Since this was a retrospective study of routinely collected data, permission from the hospital management was obtained and waiver of individual consent was granted.

## Results

Data from 319 patients were analysed, subsequently excluding 31 participants who did not meet the study criteria. These included 21 children who did not have clinical data or did not start ART before 12 years of age. One participant was known to have horizontally-acquired HIV infection, while another 9 children had non-HIV-related comorbid conditions known to affect growth (cerebral palsy (4), scoliosis (3), trisomy 21 (1) and hemiplegia (1)), leaving a total number of 288 participants, with 51.7% being male.

The majority of participants started ART late, after 6 years of age (58.3%), with a median age at ART initiation of 6.5 years (IQR 4.2;8.6) (Table 1). The average number of visits was 56 per participant (range 17 to 93). The median HAZ at ART initiation was -2.2 (IQR -3.1;-1.3), with -2.3 (IQR -3.2;-1.4) for males and -2.0 (IQR -3.1;-1.3) for females (p = 0.339). The prevalence of stunting was high at ART start (55.9%), with 29.5% severely stunted. The median baseline WAZ was -1.5 (IQR -2.5;-0.8), with 29.2% classified as underweight-for-age. The median

**Table 1. Baseline characteristics, at initiation of antiretroviral therapy, of the children living with perinatally-acquired HIV infection.**

| | | | Males | Females | Total | p-value* |
|---|---|---|---|---|---|---|
| | | | (n = 149) | (n = 139) | (n = 288) | |
| Age | | Median (IQR) | 6.6 (4.2; 8.6) | 6.4 (4.2; 8.6) | 6.5 (4.2; 8.6) | 0.817 |
| | <6 years | n (%) | 58 (38.9) | 62 (44.6) | 120 (41.7) | 0.329 |
| | ≥6 years | n (%) | 91 (61.1) | 77 (55.4) | 168 (58.3) | |
| HAZ | | Median (IQR) | -2.3 (-3.2; -1.4) | -2.0 (-3.1; -1.3) | -2.2 (-3.1; -1.3) | 0.339 |
| | Not stunted | n (%) | 60 (40.3) | 67 (48.2) | 127 (44.1) | |
| | Moderately stunted[#] | n (%) | 44 (29.5) | 32 (23.0) | 76 (26.4) | 0.328 |
| | Severely stunted[##] | n (%) | 45 (30.2) | 40 (28.8) | 85 (29.5) | |
| WAZ[&] | | Median (IQR) | -1.8 (-2.9; -0.8) | -1.3 (-2.1; -0.7) | -1.5 (-2.5; -0.8) | 0.027 |
| | Normal | n (%) | 66 (55.5) | 79 (71.8) | 145 (63.3) | |
| | Moderately UWFA | n (%) | 27 (22.7) | 17 (15.5) | 44 (19.2) | 0.035 |
| | Severe UWFA[##] | n (%) | 26 (21.8) | 14 (12.7) | 40 (17.5) | |
| BAZ[&&] | | Median (IQR) | -0.3 (-1.2; 0.6) | -0.1 (-0.9; 0.6) | -0.2 (-1.0; 0.6) | 0.287 |
| | Not wasted | n (%) | 122 (89.1) | 117 (92.9) | 239 (90.9) | |
| | Moderately wasted[#] | n (%) | 8 (5.8) | 6 (4.7) | 14 (5.3) | 0.465 |
| | Severely wasted[##] | n (%) | 7 (5.1) | 3 (2.4) | 10 (3.8) | |
| HIV Viral load | VL log | Median (IQR) | 6.1 (4.5; 5.5) | 5.0 (4.3; 5.5) | 5.0 (4.4; 5.5) | 0.364 |
| CD4 count | CD4 Abs | Median (IQR) | 400 (178; 666) | 336 (169; 644) | 357 (178; 655) | 0.685 |
| | CD4% | Median (IQR) | 13.2 (7.9; 18.8) | 13.9 (7.5; 19.1) | 13.5 (7.9; 18.9) | 0.489 |
| HIV disease severity | WHO stages 1&2 | n (%) | 9 (6.0) | 13 (9.4) | 22 (7.6) | |
| | WHO stage 3 | n (%) | 73 (49.0) | 77 (55.4) | 150 (52.1) | 0.194 |
| | WHO stage 4 | n (%) | 67 (45.0) | 49 (35.2) | 116 (40.3) | |

Abbreviations: IQR = interquartile range; HAZ = Height-for-age Z-score; WAZ = weight-for-age Z-score; BAZ = body mass index-for-age Z-score; VL = Viral load; CD4 = cluster of differentiation 4, CD4 Abs = CD4 absolute count; WHO = World Health Organization.

*p-values comparing males to females.

[#]Z-score ≤ -2 to -3 Z-score, as per WHO growth standards;

[##] Z-score ≤ -3 Z-score, as per WHO growth standards.

[&]n = 119 (males), n = 110 (females), n = 229 (total);

[&&]n = 137 (males), n = 126 (females), n = 263 (total).

baseline WAZ for males was -1.8 (IQR -2.9;-0.8), compared to -1.3 (IQR -2.1;-0.7) in females (p = 0.027). The majority of children (83%) had a normal baseline BMI for age and sex, with a median BAZ of -0.2 (IQR -1.0;0.6).

Advanced HIV disease at baseline was very common, with a median CD4% of 13.5% (IQR 7.9%;18.9%) and a median HIV VL log of 5.0 (IQR 4.4;5.5). Almost all children had WHO clinical stages 3 or 4 disease (92%), and most were started on NNRTI-based regimens (55.8%) according to the South African HIV treatment guidelines at the time.

A quarter (25%) of participants received HIV-tuberculosis co-treatment during the follow-up period according to the South African tuberculosis guidelines (Table 2).

## Growth over time

The median age at last follow-up visit was 15.8 years (IQR 14.2;17.6), with a median follow-up duration of 116.2 months (IQR 93.5;135.8) (Table 2). Most adolescents (79.9%) were no longer stunted at study exit, with a median HAZ of -1.1 (IQR -1.8;-0.6), with males having a significantly lower median HAZ (-1.4; IQR -2.2;-0.8) than females (-0.9; IQR -1.4;-0.3) (p<0.001).

**Table 2. Characteristics at study exit of adolescents living with perinatally-acquired HIV infection.**

| | | | Males | Females | Total | p-value* |
|---|---|---|---|---|---|---|
| | | | (n = 149) | (n = 139) | (n = 288) | |
| Age | Years | Median (IQR) | 15.9 (14.2; 17.8) | 15.8 (14.2; 17.6) | 15.8 (14.2; 17.6) | 0.562 |
| ART duration | Months | Median (IQR) | 116.8 (92.6;137.2) | 115.7 (94.9;134.2) | 116.2 (93.5;135.8) | 0.759 |
| HAZ | | Median (IQR) | -1.4 (-2.2; -0.8) | -0.9 (-1.4; -0.3) | -1.1 (-1.8; -0.6) | <0.001 |
| | Not stunted | n (%) | 104 (69.8) | 126 (90.7) | 230 (79.9) | |
| | Moderately stunted# | n (%) | 28 (18.8) | 12 (8.6) | 40 (13.9) | <0.001 |
| | Severely stunted## | n (%) | 17 (11.4) | 1 (0.7) | 18 (6.2) | |
| BMI | kg/m² | Median (IQR) | 18.2 (17.2; 19.7) | 19.5 (18.0; 22.4) | 18.6 (17.4; 21.1) | <0.001 |
| BAZ& | | Median (IQR) | -0.1(-1.5; -0.3) | -0.3 (-0.8; 0.6) | -0.6 (-1.4; 0.1) | <0.001 |
| | Not wasted | n (%) | 115 (91.3) | 106 (95.5) | 221 (93.3) | |
| | Moderately wasted# | n (%) | 7 (5.5) | 4 (3.6) | 11 (4.6) | 0.372 |
| | Severely wasted## | n (%) | 4 (3.2) | 1 (0.9) | 5 (2.1) | |
| Adherence to ART** | Good | n (%) | 81 (54.4) | 82 (59.0) | 163 (56.6) | |
| | Poor | n (%) | 40 (26.9) | 30 (21.6) | 70 (24.3) | 0.575 |
| | Concerning/ mixed | n (%) | 28 (18.8) | 27 (19.4) | 55 (19.1) | |
| HIV viral suppression*** | No | n (%) | 32 (21.5) | 23 (16.6) | 55 (19.1) | 0.283 |
| | Yes | n (%) | 117 (78.5) | 116 (83. 5) | 233 (80.9) | |
| TB co-infection&& | Yes | n (%) | 43 (28.9) | 29 (20.9) | 72 (25.0) | 0.117 |
| | No | n (%) | 106 (71.1) | 109 (78.4) | 215 (74.7) | |

Abbreviations: ART = antiretroviral therapy; HAZ = height-for-age Z-score; IQR = interquartile range; BAZ = body mass index-for-age Z-score; BMI = body mass index; UWFA = underweight.

*p-values comparing males to females.

**Adherence to ART: *Good* = treatment taken >90% of time. *Poor* = treatment taken <70% of time. *Concerning or mixed* = Treatment taken in 70–90% of time or included periods of good and poor treatment adherence.

***HIV viral suppression = HIV viral load <1000 copies/ml (defined as at the time of study by the World Health Organization.

#Z-score ≤ -2 to -3 Z-score, as per WHO growth standards;

## Z-score ≤ -3 Z-score, as per WHO growth standards.

&n = 126 (males), n = 111 (females), n = 237 (total);

&&n = 149 (males), n = 138 (females), n = 287 (total).

The majority of adolescents (83%) were not wasted at study exit, with a mean BAZ of -0.2 (IQR -1.0;0.6). Females had a higher BAZ at -0.1 (IQR -0.9;0.6), compared to males at -0.3 (IQR -1.2;0.6), but this was not statistically significant (p = 0.287).

Gains in WAZ and BAZ occurred during the first 36 months of ART, with minimal gains thereafter, with inability for further catch up on low baseline HAZ being especially pronounced in males (Fig 1).

All participants had at least 4 height measurements, which was then used to calculate their growth velocity during adolescence. During the pubertal growth spurt, the highest mean PHV for boys was 5.7 cm/year, attained at age 16 years of age, delayed by 24 months compared to the WHO reference population, and with an attenuated peak (Fig 2). Females had a PHV of 5.9 cm/year, attained between 12.5 and 13.5 years of age, hence delayed by 12 months, but with the pattern otherwise similar to the WHO reference population.

Children who started ART late (age ≥6 years) had delayed age at PHV (p<0.001), compared to those who started ART earlier (<6 years). The PHV and exit median HAZ in adolescents who had advanced baseline HIV disease (WHO stage 4) was lower compared to those

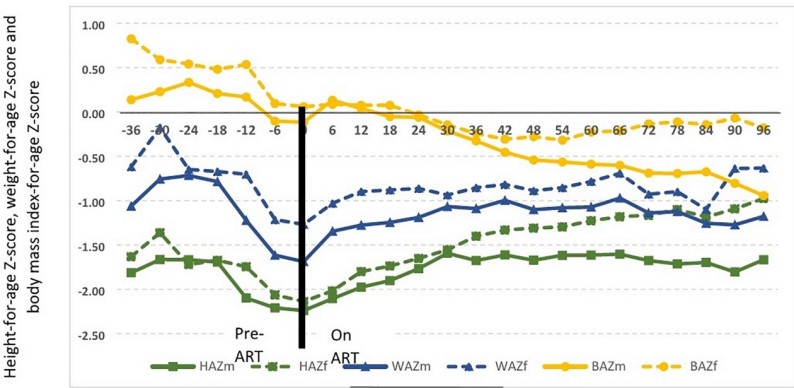

**Fig 1. Growth evolution over time before and after initiation of antiretroviral therapy using age-standardized Z-scores for height, weight and body mass index. Abbreviations**: ART = Antiretroviral therapy, HAZ = Height-for-age Z-score (HAZm = males; HAZf = females), WAZ = Weight-for-age Z-score (WAZm = males; WAZf = females), BAZ = body mass index-for-age Z-score (BAZm = males; BAZf = females).

with baseline stage 3 disease (p<0.001), but participants in both WHO stages 3 and 4 had improved their HAZ at study exit (Table 3).

Factors associated with HAZ evolution are presented in Table 4. In the descriptive model, in which HAZ was assessed at study exit, older baseline age, baseline stunting and male sex were significantly associated with a lower HAZ at study exit (all p<0.001). In the predictive models assessing final HAZ, older baseline age (p<0.001), baseline stunting (p<0.001) and male sex (p<0.007) were again significantly associated with a lower HAZ at study exit. A higher HIV VL at ART start was not significantly associated with a delay in age at PHV or a lower mean exit HAZ (p = 0.934).

## Discussion

This study documents the growth in adolescents with perinatally-acquired HIV infection in long-term care at an urban paediatric HIV clinic which was one of the first HIV treatment sites in South Africa, providing ART to public sector clients from 2004 onwards. Initially the

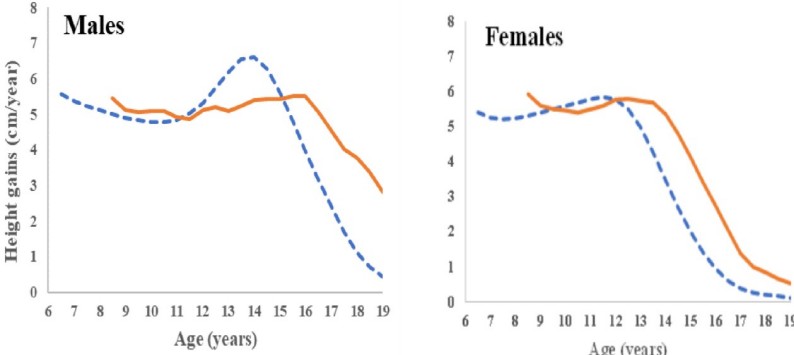

**Fig 2. Mean height gain over time (in cm/year) for males and females (orange continuous line = participants, blue dotted line = WHO reference population).**

**Table 3. Height-for-age evolution over time, stratified by clinical disease staging.**

|  |  | WHO stage 2 | WHO stage 3 | WHO stage 4 | Total |
|---|---|---|---|---|---|
| **Baseline HAZ** | Median (IQR) | -1.5 (-2.0; -1.0) | -2.0 (-2.8; -1.2) | -2.7 (-3.7; -1.9) | -2.2 (-3.1; -1.3) |
| **Exit HAZ** | Median (IQR) | -0.9 (-1.3; -0.5) | -1.1 (-1.8; -0.5) | -1.2 (-1.9; -0.6) | -1.1 (-1.8; -0.6) |
| **p-value*** |  | 0.004 | <0.001 | <0.001 | <0.001 |

Abbreviations: WHO = World Health Organization, HAZ = Height-for-age Z-score, IQR = interquartile range.

*p-value comparing WHO Baseline vs Exit was done using Median Regression on the change in HAZ.

treatment backlog was very significant, resulting in late ART initiation in many children, hence the clinical concern regarding poor linear growth and delayed pubertal growth spurt, with possible resultant negative cardiovascular, psychosocial and academic effects as well as a reduced final adult height [17–19].

Suboptimal growth was found to be a major concern in this cohort, with a marked delay in age at PHV, which was more significant in boys than in girls. These findings were similar to a

**Table 4. Generalised linear mixed effects models.**

|  |  | *Model 1* | *Model 2* |
|---|---|---|---|
|  |  | **Coefficient (95%CI)&** | **Coefficient (95%CI)&** |
| **Age at ART initiation** | <6 years | Reference- | Reference |
|  | ≥6 years | -0.29(-0.49;-0.09)* | -0.29 (-0.48;-0.10)* |
| **Stunting at baseline** | Not stunted | 1.54 (1.30; 1.78)* | 1.46 (1.23; 1.68)* |
|  | Moderately stunted# | 0.68 (0.42; 0.94)* | 0.66 (0.41; 0.91)* |
|  | Severely stunted## | Reference | Reference |
| **HIV disease severity** | WHO stages 1&2 | Reference |  |
|  | WHO stage 3 | 0.03 (-0.34; 0.40) |  |
|  | WHO stage 4 | 0.23 (-0.17; 0.62) |  |
| **Tuberculosis co-infection** | Yes | -0.07 (-0.27; 0.12) |  |
|  | No | Reference |  |
| **Adherence to ART**** | Good | Reference |  |
|  | Poor | 0.23 (-0.03; 0.50) |  |
|  | Concerning /mixed | 0.16 (-0.09; 0.42) |  |
| **Sex** | Female | 0.52 (0.34; 0.71)* | 0.51 (0.33; 0.70)* |
|  | Male | Reference | Reference |
| **Baseline HIV viral load** | Log | 0.01 (-0.06; 0.07) |  |
| **HIV viral suppression**\*** | Not suppressed | -0.16 (-0.44; 0.12) |  |
|  | Suppressed | Reference |  |

Model 1 is a descriptive model including all variables and assessing HAZ at study exit.

Model 2 is a predictive model and only included variables significantly associated with HAZ at study exit

Abbreviations: ART = antiretroviral therapy, WHO = World Health Organization, CI = Confidence interval, HAZ = Height-for-age Z-score.

&The values in the models are β coefficients (95%CI). They are the mean height-for-age Z-scores.

#Z-score ≤ -2 to -3 Z-score, as per WHO growth standards;

## Z-score ≤ -3 Z-score, as per WHO growth standards.

*Denotes a significant p-value at a 5% significance level.

**Adherence to ART: *Good* = treatment taken >90% of time. *Poor* = treatment taken <70% of time. *Concerning or mixed* = Treatment taken in 70–90% of time or included periods of good and poor treatment adherence.

***HIV viral suppression: HIV viral load <1000 copies/ml (defined as at the time of study by the World Health Organization).

study done by Jesson *et al.* [11], which also showed more normalization of growth patterns in HIV-infected girls compared to boys, which is currently not fully explained. In our study the age at PHV in girls was delayed by about 12 months and their PHV was 5.9 cm/year, while in boys the age at PHV was delayed by 24 months and the PHV was low at 5.7 cm/year, with the duration and magnitude of the pubertal growth spurt lower than the WHO reference population.

Most reported studies that illustrated an association between HIV infection and delayed pubertal onset and growth failure were done in high-income settings [4–11]. In our study, besides HIV infection itself, additional factors that were evaluated included age at ART start, HIV disease severity and tuberculosis co-infection. The mean age at ART initiation was 6.5 years. Children who started ART later (≥6 years) had a higher risk of poor growth and a delayed PHV compared to those who started earlier (<6 years), which is similar to other published studies [11, 20, 21]. There were multiple reasons for delayed ART initiation in African children, including ART guidelines current at the time, tuberculosis co-infection, lack of human resources, socio-economic obstacles and incorrect disease classification of disease severity [22, 23]. These delays in ART initiation, coupled with chronic inflammation caused by uncontrolled HIV infection, tuberculosis co-infection, recurrent opportunistic infections and poor socio-economic circumstances, all contribute to poor growth and stunting in children and adolescents living with vertically acquired HIV infection [24, 25]. The current WHO recommendation regarding early infant diagnosis and immediate ART initiation in children irrespective of their age, clinical staging or CD4 count facilitates earlier treatment access, with more studies needed to assess the catch-up growth in these newer cohorts.

Our cohort included only few children who started ART with early clinical disease (WHO stages 1 & 2). The children with WHO clinical stage 3 showed better growth compared to those with stage 4 disease at baseline. Overall, children with more advanced disease (stages 3 & 4) had improved their height at study exit but without reaching their assumed height potential (as per WHO growth standards). This is in keeping with a study done in Uganda by Bakeera-Kitaka *et al.*, in which three-quarters of children who were started on ART in WHO clinical stages 3 and 4 did not achieve their catch-up growth despite appropriate virological and immunological response to ART [26]. This emphasizes the importance of early ART initiation in children, regardless of WHO clinical staging and CD4%, as per current South African and WHO guidelines [14].

Being stunted at baseline was also associated with a delay in the age at PHV and suboptimal length growth over time. In an Asian study done by Bunupuradah *et al.*, half of the children with perinatally-acquired HIV infection who were stunted at ART initiation remained stunted over time, and a low baseline HAZ was associated with a low final height [27]. These findings are similar to our study. However, in a multi-centre, multi-country study done by the Collaborative Initiative for Paediatric HIV education and research (CIPHER), adolescents living with perinatally-acquired HIV infection in high resource income countries had a higher HAZ at study exit, compared to those living in low-and-middle-incomes countries [28]. This is in part because children in high income countries presented to care earlier and started ART at a younger age and with higher CD4 counts.

A high HIV VL at start of ART was not associated with a low median HAZ at study exit. The majority of our participants were virologically suppressed at study exit. Contrary to our findings, a study done in Southern Africa by Keiser *et al.* [29] concluded that baseline high VL was associated with poor growth, but with a small effect estimate. Good adherence to ART is usually associated with successful viral suppression and hence, disease control and a reduction in opportunistic infections, with improved childhood growth. In our study, only 57% of participants had documented good adherence to their treatment. The main barriers to good

adherence in previous studies done in sub-Saharan Africa were stigma and disclosure issues, as well as socio-economic barriers, treatment side effects and pills burden [30–33]. In our study, poor adherence to treatment was not significantly associated with a low HAZ at study exit (p = 0.330), although the documented treatment adherence may have been potentially unreliable as a clinical questionnaire and pill counts were used. Further study limitations include lack of data on pubertal developmental (e.g. Tanner staging), which would have assisted in correlating the timing of PHV. In addition, other factors, such as socio-economic and demographic data, with a potentially strong impact on the growth of adolescents living with HIV infection, were not included in this study. Despite these limitations, study strengths are that it included children with complete follow-up data for at least 5 years of treatment. The data analysis was therefore not done on repeated cross-sectional data of random children on ART, but on children in long-term care.

## Conclusion

This study demonstrated the negative impact of advanced age at ART start and the severity of baseline stunting on the pubertal growth of adolescents living with perinatally-acquired HIV infection, despite long-term ART. For reasons that require further study, boys were more affected than girls. Overall suboptimal growth in these adolescents is a significant health concern, with the possible long-term health effects becoming more important in clinical care and research as large numbers of adolescents with perinatally-acquired HIV infection are now growing into adulthood.

## Supporting information

**S1 Data.**
(XLS)

## Acknowledgments

The research team would like to acknowledge all the study participants as well as their families, all of whom have been severely impacted by the HIV epidemic in South Africa. The dedication of all the staff members working tirelessly at the Paediatric HIV Clinic at Kalafong Provincial Hospital is also acknowledged with deep respect.

## Author Contributions

**Conceptualization:** Bilema Mwambenu, Ute Feucht.

**Formal analysis:** Bilema Mwambenu, Vundli Ramoloko, Ria Laubscher, Ute Feucht.

**Investigation:** Bilema Mwambenu.

**Methodology:** Bilema Mwambenu.

**Supervision:** Ute Feucht.

**Writing – original draft:** Bilema Mwambenu.

**Writing – review & editing:** Ute Feucht.

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
