## [Decision Letter · Decision Letter 0]

25 Oct 2021

PONE-D-21-22182Growth and the pubertal growth spurt in South African adolescents living with perinatally acquired HIV infectionPLOS ONE

Dear Dr. Mwambenu,

Thank you for submitting your manuscript to PLOS ONE. After careful consideration, we feel that it has merit but does not fully meet PLOS ONE’s publication criteria as it currently stands. Therefore, we invite you to submit a revised version of the manuscript that addresses the points raised during the review process.

We look forward to receiving your revised manuscript.

Kind regards,

Evy Yunihastuti, MD

Academic Editor

PLOS ONE

Reviewers' comments:

Reviewer's Responses to Questions

**Comments to the Author**

1. Is the manuscript technically sound, and do the data support the conclusions?

Reviewer #1: Yes

Reviewer #2: Yes

2. Has the statistical analysis been performed appropriately and rigorously? 

Reviewer #1: Yes

Reviewer #2: Yes

3. Have the authors made all data underlying the findings in their manuscript fully available?

Reviewer #1: Yes

Reviewer #2: Yes

4. Is the manuscript presented in an intelligible fashion and written in standard English?

Reviewer #1: Yes

Reviewer #2: Yes

5. Review Comments to the Author

Reviewer #1: This paper is excellent and bring important data from HIV impacted country. It highlights the result of antiretroviral treatment timing on subsequent delayed growth in adolescent period.

I notice that the results were categorized according to gender of children, and showed consistent pattern in the table and graph. However in the conclusion the author included TB co-infection as one of contributing factor to growth delay. This was not supported by the GLM analysis result, model 2. Should there be other explanation for this, I think the author should revise the conclusion

Reviewer #2: This is a good information about growth in HIV adolescent with vertical transmission and described the situation in the site. The study reveals 4 times of body measurement and the authors analyzed the Growth velocity graph, is it quite enough? I think it is better make a grouping the patient's age & sex while analyzed the the growth velocity instead take all measurement to analysis; because boy and girl have different age of growth velocity.

6. PLOS authors have the option to publish the peer review history of their article (what does this mean?). If published, this will include your full peer review and any attached files.

Reviewer #1: No

Reviewer #2: **Yes: **This is a good study in high burden of HIV country and describe the HIV adolescent growth with many problems while they start their therapy

---

## [Author Response · Author response to Decision Letter 0]

21 Dec 2021

Reviewer #1: This paper is excellent and bring important data from HIV impacted country. It highlights the result of antiretroviral treatment timing on subsequent delayed growth in adolescent period.

I notice that the results were categorized according to gender of children, and showed consistent pattern in the table and graph. However in the conclusion the author included TB co-infection as one of contributing factor to growth delay. This was not supported by the GLM analysis result, model 2. Should there be other explanation for this, I think the author should revise the conclusion

Authors’ response: We thank the reviewer for this comment. We agree that our conclusion does not correlate with the results as TB infection was not significant in the adjusted GLM model 1. We have therefore removed the text relating to TB co-infection in the conclusion. 

Reviewer #2: This is a good information about growth in HIV adolescent with vertical transmission and described the situation in the site. The study reveals 4 times of body measurement and the authors analyzed the Growth velocity graph, is it quite enough? I think it is better make a grouping the patient's age & sex while analyzed the the growth velocity instead take all measurement to analysis; because boy and girl have different age of growth velocity.

Authors’ response: Thank you for the comment. The growth velocity was calculated and graphs were generated using the mean height change calculated at 6-monthly intervals. We used all data available from the time that ART was initiated, to calculate the annual change. Furthermore, we fully agree that growth in boys and girls differs; therefore all growth analyses were adjusted for child age and sex by stratification, standardization (the Z-scores) or adjustment in the regression models. The velocity analysis in figure is stratified by sex and takes age into account.

---

## [Editor Report · Decision Letter 1]

6 Jan 2022

Growth and the pubertal growth spurt in South African adolescents living with perinatally acquired HIV infection

PONE-D-21-22182R1

Dear Dr. Mwambenu

We’re pleased to inform you that your manuscript has been judged scientifically suitable for publication and will be formally accepted for publication once it meets all outstanding technical requirements.

Kind regards,

Evy Yunihastuti, MD

Academic Editor

PLOS ONE

---

## [Editor Report · Acceptance letter]

17 Jan 2022

PONE-D-21-22182R1 

Growth and the pubertal growth spurt in South African adolescents living with perinatally-acquired HIV infection 

Dear Dr. Mwambenu:

I'm pleased to inform you that your manuscript has been deemed suitable for publication in PLOS ONE. Congratulations! Your manuscript is now with our production department. 

Kind regards, 

on behalf of

Dr. Evy Yunihastuti 

Academic Editor

PLOS ONE